# Vibronic Relaxation Pathways in Molecular Spin Qubit Na$_9$[Ho(W$_5$O$_{18}$)$_2$]·35H$_2$O under Pressure

**Janice L. Musfeldt** [1,2,*] **, Zhenxian Liu** [3] **, Diego López-Alcalá** [4] **, Yan Duan** [4] **, Alejandro Gaita-Ariño** [4] **,
José J. Baldoví** [4] **and Eugenio Coronado** [4]

1    Department of Chemistry, University of Tennessee, Knoxville, TN 37996, USA
2    Department of Physics, University of Tennessee, Knoxville, TN 37996, USA
3    Department of Physics, University of Illinois Chicago, Chicago, IL 60607-7059, USA
4    Instituto de Ciencia Molecular, Universitat de Valencia, 46980 Paterna, Spain
*    Correspondence: musfeldt@utk.edu

**Abstract:** In order to explore how spectral sparsity and vibronic decoherence pathways can be controlled in a model qubit system with atomic clock transitions, we combined diamond anvil cell techniques with synchrotron-based far infrared spectroscopy and first-principles calculations to reveal the vibrational response of Na$_9$[Ho(W$_5$O$_{18}$)$_2$]·35H$_2$O under compression. Because the hole in the phonon density of states acts to reduce the overlap between the phonons and $f$ manifold excitations in this system, we postulated that pressure might move the HoO$_4$ rocking, bending, and asymmetric stretching modes that couple with the $M_J = \pm5$, $\pm2$, and $\pm7$ levels out of resonance, reducing their interactions and minimizing decoherence processes, while a potentially beneficial strategy for some molecular qubits, pressure slightly hardens the phonons in Na$_9$[Ho(W$_5$O$_{18}$)$_2$]·35H$_2$O and systematically fills in the transparency window in the phonon response. The net result is that the vibrational spectrum becomes less sparse and the overlap with the various $M_J$ levels of the Ho$^{3+}$ ion actually increases. These findings suggest that negative pressure, achieved using chemical means or elongational strain, could further open the transparency window in this rare earth-containing spin qubit system, thus paving the way for the use of device surfaces and interface elongational/compressive strains to better manage decoherence pathways.

**Keywords:** spin qubit; vibronic coupling; strategies to minimize decoherence; high pressure vibrational spectroscopy





## 1. Introduction

Single molecule magnets incorporating heavy centers are of foundational importance for exploring orbital localization and chemical bonding, electron correlation vs. spin–orbit coupling, and the different patterns of crystal field energy levels [1–5]. Hundreds have been developed in an effort to control the properties and reveal structure–property relations [6–9]. Several have demonstrated spin qubit behavior [10–14]. Prominent examples include (i) (PPh$_4$)$_2$[Cu(mnt)$_2$] (mnt$^{2-}$ = maleonitriledithiolate) doped into the diamagnetic isostructural host (PPh$_4$)$_2$ [Ni(mnt)$_2$], (ii) [(Cp$^{iPr5}$)Dy(Cp*)]$^+$ (where Cp$^{iPr5}$ = penta-iso-propylcyclopentadienyl and Cp* = pentamethylcyclopentadienyl), and (iii) chiral [Zn(OAc)(L)Yb(NO$_3$)$_2$)] as well as many others [15–18]. In the Ln$^{3+}$-containing family of mononuclear molecular nanomagnets, Na$_9$[Ho(W$_5$O$_{18}$)$_2$]·35H$_2$O is attracting considerable attention [15,19–21]. This is because of the 8.4 μs coherence time in diluted systems [20] as well as the fact that the spin qubit dynamics are protected against magnetic noise at favorable operating points known as atomic clock transitions [14,20,22], where this system has also been experimentally found to display magnetoelectric coupling [23]. Of course, even with clock protection, quantum information can be lost through vibrational and thermal processes [4,17,24–30], although very few of these mechanisms have been unraveled in a detailed manner [31–37].

To address this issue, our team recently began exploring decoherence pathways in $Na_9[Ho(W_5O_{18})_2]\cdot35H_2O$ with the goal of revealing specific vibronic relaxation pathways governing magnetic relaxations [38]. We discovered strong magneto-infrared contrast near 370 and 63 $cm^{-1}$ due to mixing of odd-symmetry vibrations with $f$-manifold crystal field excitations. Specifically, the $M_J = \pm7$ crystal field levels couple to the various $HoO_4$ rocking and bending modes. At the same time, the $M_J = \pm5$ levels near 63 $cm^{-1}$ (and very likely the $M_J = \pm2$ levels) are activated by nearby phonons such as asymmetric $HoO_8$ stretching with cage tilting. Moreover, we reported the first direct evidence for a transparency window in the phonon density of states in a robust clock-like molecular spin qubit. The overall extent of vibronic coupling [17,26–28] in $Na_9[Ho(W_5O_{18})_2]\cdot35H_2O$ is therefore limited by a modest coupling constant (in the order of 0.25 $cm^{-1}$) and a transparency window in the phonon density of states that acts to keep the intramolecular vibrations and $M_J$ levels apart [38]. This is different from $3d$-containing molecule-based materials such as $Co[N(CN)_2]_2$ where significantly larger spin–phonon coupling constants (2 and 3 $cm^{-1}$) give rise to avoided crossings and a transfer of oscillator strength from nearby phonons to the localized $Co^{2+}$ electronic excitations and back again under magnetic field [39]. Despite the smaller coupling constant, interaction with phonons is still a significant problem in $Na_9[Ho(W_5O_{18})_2]\cdot35H_2O$—even at low temperature. Recent simulations in entangled two-qubit gates suggest that increased spin–lattice relaxation time ($T_1$) is likely with additional cooling [30]. Since in this system $T_2$ is controlled by $T_1$, this is relevant for its behavior as a qubit.

Decoherence of quantum states in a qubit can occur when resonances of different types are found in close proximity. The natural spectral sparsity in $Na_9[Ho(W_5O_{18})_2]\cdot35H_2O$ raises questions about whether even more extreme separations between the magnetic and vibrational excitations can be encouraged and even promoted [40–43]. Besides exploring the consequences of this "hole" on vibronic coupling, we recently proposed several design rules aimed at mitigating decoherence pathways involving vibronic coupling [38]. In addition to isotope effects and studies of chemically analogous $Ho^{3+}$-containing polyoxometalates, we discussed specific suggestions for alternate rare earth ions and coordination effects as well as stiffer surrounding ligands [4,38]. One promising avenue that was not explored in prior work is the effect of pressure. Compression changes bond lengths and angles [44] and, as a result, tends to harden phonons—perhaps moving them out of the way. Such an approach has the potential to significantly reduce vibronic coupling as a decoherence mechanism by moving these excitations off resonance [29,38,40–42,45].

In order to to test whether pressure can disentangle electronic and vibrational excitations in a molecular magnet with atomic clock transitions, we measured the far infrared response of $Na_9[Ho(W_5O_{18})_2]\cdot35H_2O$ under compression and compared our findings with complementary first-principles calculations. This work is based upon the premise that both the molecular vibrations and the $M_J$ levels that derive from on-site $f$-manifold excitations of $Ho^{3+}$ are likely to harden under compression and therefore might be able to be shifted to nullify the spin–spin ($T_2$) relaxation pathway under small pressures or strain. In other words, pressure is expected to act upon both energy scales, in the same direction (and in both cases, the desired direction), although not necessarily to the same extent. A more compressed molecule will be more rigid, but also closer metal–ligand distances will result in a stronger crystal field. This is what one wants: molecules that are very rigid and with very strong crystal field perturbations. This energy scaling should, in principle, systematically decrease the chance of a close resonance. Rendering the $M_J$ levels ineffective in terms of engaging in vibronic coupling, we find that pressure broadens the low frequency phonons and shifts others more strongly into resonance. At the same time, compression works to systematically fill the hole in the phonon density of states, so rather than making a sparse lattice even more sparse, pressure decreases sparsity by closing the transparency window. These findings, supported by our theoretical calculations, suggest that negative pressures, obtained via crystal engineering or elongational strain, will be more effective to enhance the performance of this molecular spin qubit.

## 2. Methods

### 2.1. Experimental Setup

High quality $Na_9[Ho(W_5O_{18})_2]\cdot35H_2O$ single crystals were grown as described previously [20]. A small, well-shaped piece was selected and loaded into a suitably chosen diamond anvil cell with an annealed ruby ball and hydrocarbon grease (petroleum jelly) as the pressure medium in order to assure quasi-hydrostatic pressure conditions (Figure 1a). In addition to using the ruby ball to determine pressure via fluorescence [46,47], we monitored the shape of the ruby fluorescence spectrum to assure that the sample remained in a quasi-hydrostatic environment (Figure 1b). The synthetic type IIas diamonds in the symmetric diamond anvil cell had 500 μm culets and we employed a 47 μm thick preindented stainless steel gasket with a 200 μm hole diameter. Care was taken to optimize optical density in order to reveal the features of interest. In fact, we carried out two different trials using both low and high optical densities. We employed high optical density in the 55–370 $cm^{-1}$ range where the features have smaller intensity, and we used lower optical density in the 300–625 $cm^{-1}$ range where the spectral features are stronger. Taking advantage of the stable, high-brightness beam, synchrotron-based infrared spectroscopy (60–680 $cm^{-1}$; 4 $cm^{-1}$ resolution; transmittance geometry) was performed using the 22-IR-1 beamline at the National Synchrotron Light Source II at Brookhaven National Laboratory. Absorbance is calculated as $\alpha(\omega) = -\ln(\mathcal{T}(\omega))$, where $\mathcal{T}(\omega)$ is the measured transmittance. Pressure was increased between 0 and 5.2 GPa at room temperature. Our prior work revealed no substantial spectral changes in the phonons or the transparency window with temperature down to 9 K [38], so we did not pursue these effects here. The spectral changes are fully reversible upon release of pressure.

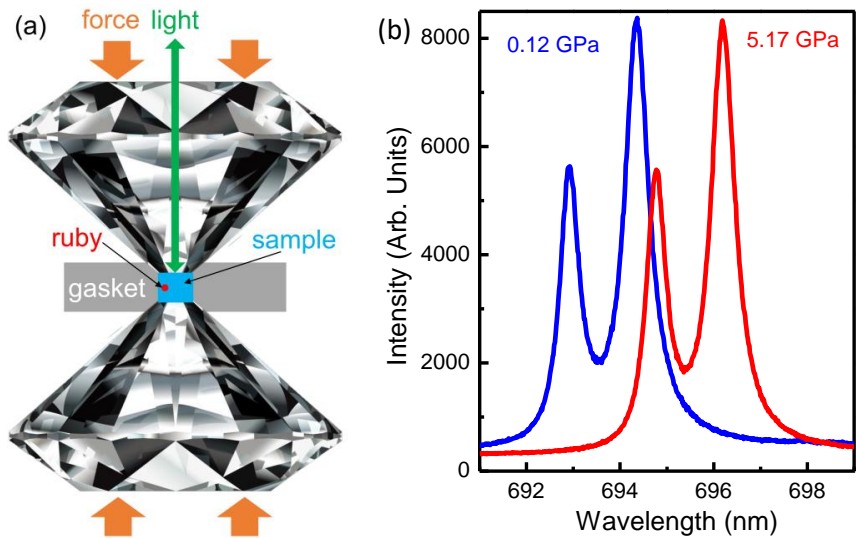

**Figure 1.** (**a**) Schematic of the symmetric diamond anvil cell used in these measurements. The ruby ball and the sample crystal share a high pressure environment but are not directly in contact. (**b**) Fluorescence of the annealed ruby ball when the cell was just closed (0.12 GPa) and at 5.17 GPa. The shape of the fluorescence spectra along with the separation of the two peaks is consistent with quasi-hydrostatic conditions.

### 2.2. DFT Calculations

The optimization of the molecular geometries and the simulation of vibrational spectra were carried out using the Gaussian09 package in its revision D01 [48]. The PBE0 hybrid-exchange correlation functional was used in all the calculations [49]. Different basis sets were used for each type of atom in the molecule: Stuttgart RSC ANO basis set [50–52] for the $Ho^{+3}$ cation, CRENBL basis set [53] for W, and 6-31G** for O [54,55]. Their corresponding effective core potentials (ECP) for Ho and W atoms were applied. We included Grimme D3BJ dispersion corrections in all the calculations [56]. First, we optimized the

crystal structure and then we determined the vibrational spectra. We optimized all the structures until the change in energy of two consecutive steps was lower than $10^{-6}$ Hartree. The threshold assigned for maximum forces and RMS matrix was $4.5 \cdot 10^{-4}$ and $3 \cdot 10^{-4}$ Hartree/Bohr, respectively. Then, we applied triaxial pressure by decreasing each Cartesian coordinate by 0.5, 1, 1.5, and 2%. This allowed us to simulate the effect of mechanical pressure. Finally, we performed a constrained optimization to maintain the effect of the pressure, thus avoiding negative vibrational frequencies in the spectrum. This constraint consisted of freezing the position of the atoms lying in the Cartesian axes, thus keeping the effect of the triaxial pressure.

### 2.3. Semi-Empirical Crystal-Field Calculations

To calculate the effect of the applied strain on the energy level scheme of the ground-J manifold, we applied the semi-empirical radial effective charge (REC) model [57], as implemented in the SIMPRE computational package [58]. In order to account for covalency effects, we applied a radial displacement of ($D_r$ = 0.48 Å) and effective point-charges ($Z_i$ = 0.81). These parameters were extracted by fitting the spectroscopic energy levels determined by Vonci et al. [21].

### 3. Results and Discussion

Figure 2 summarizes the far infrared response of $Na_9[Ho(W_5O_{18})_2] \cdot 35H_2O$ as a function of pressure at room temperature. The main difference between the two trials in panels (a) and (b) is the optical density, which allows us to examine different spectral regions with optimum sensitivity. Molecular materials are well known to be soft and flexible but, unlike a number of other systems [59–61], there is no evidence for a structural phase transition between 0 and 5 GPa in $Na_9[Ho(W_5O_{18})_2] \cdot 35H_2O$. Instead, compression acts to (i) harden the vibrational modes and (ii) reduce spectral sparsity. These findings are discussed below.

As we know, vibrations play an important role in magnetic relaxation processes of molecular spin qubits as they couple to spin states, leading to the loss of quantum information [17,26–28]. In $Na_9[Ho(W_5O_{18})_2] \cdot 35H_2O$, we are primarily interested in vibronic decoherence pathways involving odd-symmetry vibrations near 370 and 63 cm$^{-1}$ that mix with $f$-manifold crystal field excitations [38]. As expected, pressure hardens the majority of vibrational modes at a rate of approximately 0.9 cm$^{-1}$/GPa (Figure 2a,c), so it is possible to push a mode that is on resonance and therefore detrimental (such as those near 370 or 63 cm$^{-1}$) away from resonance with a particular $M_J$ level of the Ho$^{3+}$ ion. In this system, however, the vibrational modes do not harden at significantly different rates so, rather than taking a particular mode off-resonance and leaving the rest unperturbed, the full set of modes hardens systematically. Above approximately 1 GPa, these modes begin to interact with other spin levels in $Na_9[Ho(W_5O_{18})_2] \cdot 35H_2O$—not just the $M_J = \pm 7, \pm 5$, and $\pm 2$ levels—so it is really only the smallest pressure (or presumably strain) that is potentially useful.

Our prior analysis suggests that coherence in $Na_9[Ho(W_5O_{18})_2] \cdot 35H_2O$ benefits from the limited frequency overlap between Ho$^{3+}$ crystal field levels and the phonon manifold [38]. The limited overlap is due to a transparency window or "hole" in the phonon density of states that renders many of the $M_J$ levels ineffective in terms of engaging in vibronic coupling. Revealing how pressure affects the transparency window in the phonon density of states is therefore extremely important. For this technique to work well, the hole should stay the same size or expand slightly. Unfortunately, the transparency window in the phonon response closes systematically under compression (Figure 2a). We quantify this effect by integrating the area under the absorption in this frequency window and plotting it as a function of pressure (inset, Figure 2a). Initially, this quantity grows linearly with pressure. It levels off around 5 GPa as the filling saturates and the transparency window closes. Therefore, while the tiniest bit of compression might impact the coherence time in a positive manner, in general, pressure is not an effective external stimulus because it elimi-

nates the transparency window in the vibrational spectrum. Broadening the transparency window with "negative pressure" would be a better approach—at least in this system.

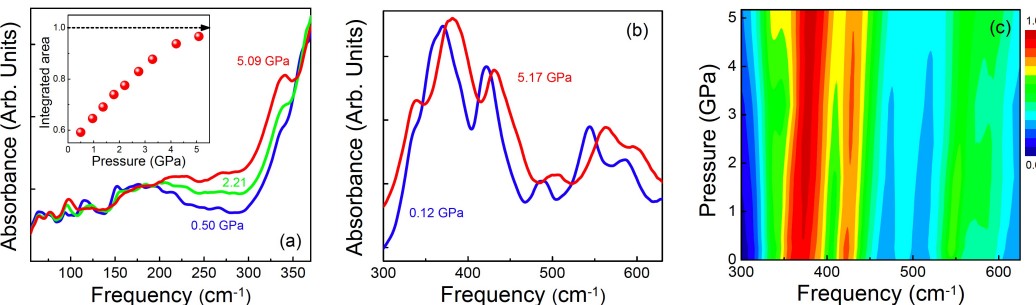

**Figure 2.** (**a**) Synchrotron-based far infrared spectrum of $Na_9[Ho(W_5O_{18})_2]\cdot35H_2O$ as a function of pressure in the 55–375 $cm^{-1}$ region. Both intramolecular and intermolecular features are observed. Inset: Integrated area under the curve in the 220–310 $cm^{-1}$ frequency window. The latter corresponds to the transparency window and was therefore used as the integration range. (**b**) Close-up view of the far infrared response of $Na_9[Ho(W_5O_{18})_2]\cdot35H_2O$ between 300 and 625 $cm^{-1}$. (**c**) The same data as in panel (**b**) shown as a contour plot. Spectra at 12 different pressures were used to create the contour plot.

In order to rationalize these findings, we performed first-principles calculations on a fully optimized $[Ho(W_5O_{18})_2]^{9-}$ molecule at the density functional theory level. We applied different compressive triaxial strain values ranging from 0.5 to 2% (Figure 3a), which are compatible with the structural changes of the crystallographic coordinates in the experiment, assuming a typical Young's modulus of ~100 GPa for this class of crystals. Our results corroborate the hardening of the molecular vibrations under compression, thus supporting the trend observed in the synchrotron-based far infrared measurements (Figure 2b). This is expected due to the shorter bond lengths in the molecule, which are displaced from their equilibrium positions, thus enhancing the force constant of the bonds within the harmonic oscillator approximation. At the same time, we computed the evolution of the $M_J$ energy level scheme under the same compressive strain values (Figure 2b). As one can observe, the reduction of the bond distances between the $Ho^{3+}$ and the coordinated oxygen centers leads to a linear increase in the crystal field splitting, considering covalency effects by an effective displacement of the electrostatic point charges, under strain. This linear response of the crystal field is the expected behavior at small distortions. The crystal field terms respond to changes in metal–ligand distance as $O_k^q \propto 1/R^{(k+1)}$ but for distortions in the order of 2% these are very well approximated by linear behavior. In Figures S1–S5 of the Supporting Information, we display a combination of both calculated crystal field energy levels and the infrared spectrum for each applied strain.

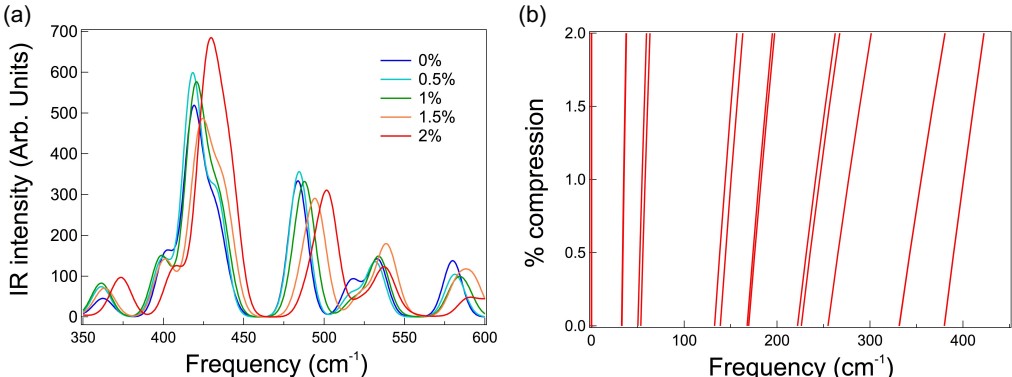

**Figure 3.** (**a**) Calculated infrared spectra of the building block anion $[Ho(W_5O_{18})_2]^{9-}$ at different compressive strain values (from 0 to 2%) in the 350–600 cm$^{-1}$ region. (**b**) Evolution of the spin energy levels of $Ho(W_5O_{18})_2]^{9-}$ under compressive strain calculated by the radial effective charge model.

$Na_9[Ho(W_5O_{18})_2]\cdot 35H_2O$ already benefits from a relatively sparse lattice (due to the hole in the phonon density of states), and it is likely that it can be made sparser with "chemical pressure" or elongational strain. In our target system, larger more covalent counterions might encourage overall lattice expansion or even just local relaxation sufficient to expand the hole in the phonon density of states. Crystal engineering with chemical pressure is very much akin to chemical control of a host matrix environment [29,38,40–43,45]. At the same time, elongational strain can be tested using a piezostack and incorporated into a device using well-known principles of strain engineering and substrate choice. Again, the idea would be to shield the $M_J$ levels of $Ho^{3+}$ by moving the phonons that engage in the strongest coupling out of the way—without introducing new opportunities for interaction. It is far from certain that the clock transitions will remain near 9 GHz (the X-band) under pressure or strain. Small distortions have already been predicted to change the energy of the clock transitions, indicating the fragility of this parameter [22]. Theory can also test strategies for expanding the lattice and blocking decoherence pathways, simulating the effect of the modified phonon density of states on the vibronic coupling constants and the $T_1$ relaxation time. This approach to tuning the relative position of states, if successful, has the potential to accelerate the development of molecular spin qubits with improved lifetimes, electric field control, and higher operating temperatures [4,23].

## 4. Summary and Outlook

$Na_9[Ho(W_5O_{18})_2]\cdot 35H_2O$ is a model spin qubit system with atomic clock transitions and a relatively sparse vibrational spectrum that supports limited overlap between the $f$ manifold levels of $Ho^{3+}$ and the infrared-active vibrational modes, circumstances that favor modest vibronic coupling constants and longer coherence times. In this work, we tested whether an external stimulus in the form of pressure could manipulate this favorable situation even further, perhaps expanding the hole in the phonon density of states to reduce the overlap of these states (and thus the importance of vibronic decoherence pathways) still further. To test this strategy, we measured the far infrared response of $Na_9[Ho(W_5O_{18})_2]\cdot 35H_2O$ under pressure. It turns out that the vibrational modes of this molecular nano-magnet harden under compression. This is in agreement with our simulation of the far infrared spectrum of the evolution of molecular vibrations under compressive strain. On the other hand, the transparency window in the phonon density of states also begins to close, being filled by approximately 5 GPa. These findings suggest that rare earth-containing molecular spin qubits like $Na_9[Ho(W_5O_{18})_2]\cdot 35H_2O$ would instead benefit from negative pressure. While there are a number of efforts to tune similar systems using chemical pressure, tensile strain is under-explored as a technique for disentangling these processes. In this context, device surfaces and interfaces may offer important opportunities in future work.

**Supplementary Materials:** The following supporting information can be downloaded at: https://www.mdpi.com/article/10.3390/magnetochemistry9020053/s1.

**Author Contributions:** J.L.M. conceived the study and carried out the high pressure synchrotron far infrared measurements in collaboration with Z.L.; and J.L.M. treated the data, drew the figures, and drafted the manuscript. D.L.-A. performed the theoretical simulations and corresponding figures supervised by J.J.B.; A.G.-A. contributed to the theoretical analysis. Y.D. synthesized the sample under the supervision of E.C. All authors have read and agreed to the published version of the manuscript.

**Funding:** It is our pleasure to contribute this article to the issue of *Magnetochemistry* in honor of Professor Manuel Almeida on the occasion of their retirement. Thanks for so many useful discussions. Research at the University of Tennessee is supported by the National Science Foundation (DMR-1707846). Work at the National Synchrotron Light Source II at Brookhaven National Laboratory is funded by the Department of Energy (DE-AC98-06CH10886). Use of the 22-IR-1 beamline is supported by COMPRES, the Consortium for Materials Properties Research in Earth Sciences, under NSF Cooperative Agreement EAR 1606856 and CDAC (DE-NA0003975). Research at Universitat de Valencia is supported by the EU (ERC-2018-AdG-788222 MOL-2D) and the QUANTERA project SUMO; the Spanish MCIU (grant CTQ2017-89993 and PGC2018-099568-B-I00 cofinanced by FEDER, grant MAT2017-89528; the Unit of excellence 'María de Maeztu' CEX2019-000919-M); and the Generalitat Valenciana (Prometeo Program of Excellence, SEJI/2018/035 and grant CDEIGENT/2019/022).

**Institutional Review Board Statement:** None.

**Informed Consent Statement:** None.

**Data Availability Statement:** Data are available from the corresponding author upon reasonable request.

**Acknowledgments:** We thank A. Ullah for useful conversations.

**Conflicts of Interest:** The authors declare no conflict of interest.

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
