# Peer review of "Vibronic Relaxation Pathways in Molecular Spin Qubit Na9[Ho(W5O18)2]·35H2O under Pressure"

_magnetochemistry, doi:10.3390/magnetochemistry9020053_

Round 1
Reviewer 1 Report
The search for novel materials for quantum technologies lies in the core of modern physics and chemistry of magnetic materials. The manuscript reports an experimental and theoretical study of energy levels of Na_9[Ho(W_5O_18)_2]*35H_2O molecule by means of far-infrared spectroscopy under high pressure and Density Functional Theory calculations. The most crucial finding of this work is that compressive strain tends to close the transparency gap in the infrared spectrum, therefore, tensile strain would exert an opposite effect, improving the suitability of the molecule for the usage in quantum information processing. The paper is well written and reports interesting and novel results of interest to the community.
I recommend it for publication in Magnetochemistry journal, just after the Authors consider the minor points listed below:
* I think that some scheme of the molecule of interest would be optionally provided, for example to illustrate the output of structural relaxation within DFT approach.
* Page 3, line 116: I believe that giving some more details on the procedure of relaxation (for example force threshold) would be of interest.
* Page 3, line 119: it might be interesting to explain in a little more detailed way what sort of constraints was used in progress of optimization.
* Page 4, Fig. 2: a technical problem - the figure is too wide. In its caption, the explanation of the inset would include the information that the range of integration corresponds to the transparency window.
* Supplementary, page 2, Fig. 1. In the caption, the information that the data were calculated in the absence of strain would be added for full clarity.
* Supplementary: Numeration of figures should be, I guess, S1 to S5 (as referenced in the main text).
Reviewer 2 Report
This manuscript reports on a preliminary investigation, theoretical and experimental, of the effect of pressure on the far infrared response of the lanthanide based qubit Na9[Ho(W5O18)2]35H2O under compression. The underlying idea was that of testing whether this approach, resulting in a more rigid molecule with stronger crystal field has the potential to reduce vibronic coupling by moving phonons and energy levels off-resonance. However, both the experimental results and the theoretical analysis show that while the vibrational modes harden under compression, the transparency window in the phonon DoS is narrowed on increasing pressure. Consequently, it is suggested that applying a “negative pressure” would be a more appropriate strategy to reduce vibrationally induced decoherence.
The subject of the paper is for sure interesting for Magnetochemistry, and the results, albeit preliminary, are worth publishing after some revisions, in the scholar presentation form and reference section. The suggested changes are detailed below:
1- In the Introduction, three prominent examples are given: however, the corresponding references 15-18 do not correspond to these. (i) is indeed reported in ref. 10; ref. 15 does not correspond to any of (i)-(iii). Reference 31 is called before 23. Some more accurate ordering of the references would then be welcome.
2- Despite the paper being centered on the effect of vibronic coupling on the spin relaxation of molecular qubits, references to literature in this field are extremely limited, essentially to authors themselves. As an example, the framing of this work in the context of previous works such as those of Chilton’s group (eg Nature Communications 2022), Sessoli’s group (eg Inorg. Chem 2021) and – from a theoretical point view - Lunghi’s group is lacking. At the same time, inclusions of some of coauthors’ references, eg ref. 28, is irrelevant for the research presented here, and might even represent inappropriate self-citation.
3- L. 52 and 72, T2 (spin-spin) and T1 (spin-lattice) are swapped.
4- L.134-135 anticipates the interpretation of results which have not been yet presented. These two lines should be moved at least after l. 164.
5- Fig. 2 has some formatting issues (the rightmost part of panel (c) is cutted).
6- In the caption to fig 2 c, the number of experimental points upon which the graph is obtained should be written explicitly. In the current version it appears almost as a continuum of pressures being investigated, whereas according to inset of Fig. 2(a) only a few pressures have been investigated.
7- L. 136-140 contain a concept already presented in the Introduction, there is no need to repeat it here.
8- L. 142 “offending” is not easy to understand here. Do they intend “detrimental for relaxation”?
9- L.177-178. The way it stands now, the sentence on these lines seems to imply that covalence effects are not considered, the dependence of the CF operators on distances being the one expected for purely electrostatic interactions. For reader less aware about the working principles of SIMPRE, a few words should be spent about the way it models covalency.
Reviewer 3 Report
The paper discusses the interplay between magnetic relaxation and spin decoherence of the single molecule magnets Na9[Ho(W5O18)2]·35H2O and molecular vibrations or phonons. In particular, the effects of hydrostatic pressure is studied systematically here. The paper is well written, the experiments seem to be well done and the analysis is good. Therefore, I can recommend publication of this work in Magnetochemistry once the authors address the following minor issues,
- Line 50: should be "still a significant problem"
- Line 52: "suggest that improvements to the spin-lattice relaxation time (T2) are likely with additional cooling.". It is not clear what "improvement" means in this context, I guess it is longer spin-lattice relaxation but the authos should clarify this in the text. Also, in NMR, spin lattice relaxation is referred to as T1 not T2.
- Line 73: Also here, spin-spin relaxation is T2 not T1.
- Line 79: "instead of rending the M_J ...", I think that it should be "rendering"
- Figure 2(c) is clipped on the right side, part of the figure is missing.
Line 162: The authors talk about "transparency window" and it is not clear to me where this window is, maybe the authors can highlight that with a shaded area in the figures.
- Caption of Fig. 3(a): "infrared spectrum" should be "infrared spectra" for plural.
